# Large-Scale Sequencing of *Borreliaceae* for the Construction of Pan-Genomic-Based Diagnostics

**DOI:** 10.3390/genes13091604

**Published:** 2022-09-08

**Authors:** Kayla M. Socarras, Benjamin S. Haslund-Gourley, Nicholas A. Cramer, Mary Ann Comunale, Richard T. Marconi, Garth D. Ehrlich

**Affiliations:** 1Center for Advanced Microbial Processing, Institute for Molecular Medicine and Infectious Disease, Drexel University College of Medicine, Philadelphia, PA 19102, USA; 2Center for Genomic Sciences, Institute for Molecular Medicine and Infectious Disease, Drexel University College of Medicine, Philadelphia, PA 19102, USA; 3Department of Microbiology and Immunology, Drexel University College of Medicine, Philadelphia, PA 19102, USA; 4Department of Microbiology and Immunology, Virginia Commonwealth University Medical Center, 1112 East Clay Street, Room 101 Health Sciences Research Building, Richmond, VA 23298, USA; 5Department of Oral and Craniofacial Molecular Biology, Philips Institute for Oral Health Research, School of Dentistry, Virginia Commonwealth University, Richmond, VA 23298, USA; 6Center for Surgical Infections and Biofilms, Institute for Molecular Medicine and Infectious Disease, Drexel University College of Medicine, Philadelphia, PA 19102, USA

**Keywords:** tick-borne diseases, Lyme disease, borrelia, pangenomics, diagnostics, distributed genome hypothesis

## Abstract

The acceleration of climate change has been associated with an alarming increase in the prevalence and geographic range of tick-borne diseases (TBD), many of which have severe and long-lasting effects—particularly when treatment is delayed principally due to inadequate diagnostics and lack of physician suspicion. Moreover, there is a paucity of treatment options for many TBDs that are complicated by diagnostic limitations for correctly identifying the offending pathogens. This review will focus on the biology, disease pathology, and detection methodologies used for the *Borreliaceae* family which includes the Lyme disease agent *Borreliella burgdorferi*. Previous work revealed that *Borreliaceae* genomes differ from most bacteria in that they are composed of large numbers of replicons, both linear and circular, with the main chromosome being the linear with telomeric-like termini. While these findings are novel, additional gene-specific analyses of each class of these multiple replicons are needed to better understand their respective roles in metabolism and pathogenesis of these enigmatic spirochetes. Historically, such studies were challenging due to a dearth of both analytic tools and a sufficient number of high-fidelity genomes among the various taxa within this family as a whole to provide for discriminative and functional genomic studies. Recent advances in long-read whole-genome sequencing, comparative genomics, and machine-learning have provided the tools to better understand the fundamental biology and phylogeny of these genomically-complex pathogens while also providing the data for the development of improved diagnostics and therapeutics.

## 1. Introduction

Even in this era of modern medicine, including mass vaccination achievements and antibiotic treatment regimens, vector-borne diseases such as Lyme borreliosis, Malaria, Dengue fever, yellow fever, and bubonic plague still prevail. This group of ancient and persistent diseases is transmitted to humans through the bite of infected arthropod vectors including mosquitoes, lice, flies, and ticks. These vectors acquire their associated pathogens through blood feedings from multiple hosts throughout complex life cycles. This life-cycle complexity leads to difficulties in diagnoses and treatments, creating a persistent healthcare burden that has resulted in a push to identify biomarkers for the development of improved diagnostics, vaccines, and therapeutics [1].

Research on these fronts has been divided unevenly across different vector species. While a great deal of research has been performed on diseases transmitted by mosquitoes, tick-borne diseases (TBD) are less studied. As their name implies, TBDs are primarily transmitted through the bite of infected ticks. While these small ectoparasitic arachnids have long been documented to cause disease, some as early as 1550 B.C., their true clinical significance was not realized until 1893, when a publication by Smith and Killborne linked *Rhipicephalus annaltus* to the transmission of the protist parasite *Babesia bigemina* in cattle [2,3,4,5,6]. Despite this finding, research on ticks and their corresponding TBDs progressed slowly until the latter half of the 20th century.

Moreover, research in the field progressed on TBDs unevenly, often centering on the most prevalent pathogen (or what was thought to be the most prevalent pathogen) or vector in a given region rather than the wide array of TBD pathogens and their vectors. This is most clearly seen with Lyme borreliosis (Lyme disease), caused by the *Borreliella burgdorferi sensu lato* spirochetes and their known vectors, *Ixodes scapularis* and *I. pacificus*, within the Northeastern and Pacific regions of the United States (US), respectively. Research efforts initially centered on the mammalian immune response towards these spirochetes, leading to developments of first-generation diagnostics that have some efficacy but are fraught with poor sensitivity [7,8]. While this work is important, more research is needed to improve the efficacy of Lyme borreliosis diagnostics, including biomarker discovery, nucleic-acid-based methods, and improved serological methods. This review will cover the TBD field as it stands, with emphasis on borreliosis, specifically Lyme borreliosis, and outline plausible paths for the development of better diagnostics and therapeutics.

## 2. The Macro and Micro Ecology of Ticks: Their Role in the Environment and Their Microbiomes

The *Agrasidae* have 186 recognized species and are commonly referred to as ‘soft’ ticks due to the absence of a scutum, a hard protective plate, and the exposure of their mouths [9]. The second, larger *Ixodidae* tick family has 720 recognized species. These arachnids are commonly known as ‘hard’ ticks due to the presence of their scutum (Figure 1). While both families have distinct morphologies, all can thrive in various geographic niches ranging from the tropics to the subarctic [10]. Though many tick species are found in a range of climates, most tick species are concentrated in tropical/subtropical regions [10].

Recently, due mainly to climate change, the geographic range of many tick species has expanded poleward beyond their traditional habitats [11,12]. This restructuring has resulted from a variety of reasons including prey migration, local environmental degradation, and increased temperatures associated with global warming [11,12,13,14]. The latter facilitates increased tick survival through winters within previously uninhabitable regions [15,16,17,18,19,20]. Evidence of this change is seen in numerous tick species. A tick species that is able to live in both hemispheres and extreme climates is *Ixodes uriae*, a seabird tick [21,22] that is now prevalent in Nordic Countries and can transmit *Borrelia burgdorferi*, *Borrelellia garinii*, and *Borrelia bavariensis* [23,24,25]. Similarly, *Haemaphysalis longicornis*, the Asian Longhorn tick, has recently arrived in the US [26,27]. This invasive species has rapidly integrated within its new ecosystem and has joined the list of clinically relevant tick species found within the US (Table 1) [26,28]. Additionally, several native North American tick species are also migrating further north. Some have been documented to parasitize wildlife in the harsh frontiers of Alaska [29,30].

As the habitat zones of ticks change, so does the epidemiology of the pathogens they transmit [2]. This review will focus on the most significant tick species that carry human pathogens (Table 1) including *I. scapularis* (black-legged tick) and its west-coast sister species *I. pacifius* (western black-legged tick); *Dermacentor variabilis* (American dog tick); and *Amblyomma americanum* (Lone star tick).

The most well-known of these is *I. scapularis.* This species is fascinating for its biology and its role in medical history. The clinical significance of the deer tick was realized when it was linked with outbreaks of a rheumatoid arthritis-like disease in children in Lyme, Connecticut [31]. This mysterious outbreak lasted for years and went well into the onset of the AIDS epidemic. It was originally postulated as a viral disease [32,33]. In the years that followed, this was disproven when a ‘treponema-like’ spirochete was isolated from the blood of local fauna and afflicted individuals [34,35,36,37]. This spirochete was named *Borrelia burgdorferi* in honor of the lead scientist Dr. Wilhelm Burgdorfer [38]. In the years that followed, the maladies caused by *B. burgdorferi* were called Lyme disease, now referred to as Lyme borreliosis (Lb) to specify the clinical infection [36,37]. Throughout the remainder of this review, Lb shall be the predominant term used.

Despite identifying the Lb etiological agent, its true vector remained unknown. Early investigations for the Lyme borreliosis vector focused on the prevalent New England tick species known to transmit pathogens including *D. variabilis* and *Ixodes dammini.* The latter *Ixodes* species was first hypothesized as the host of *B. burgdorferi*—a concept later proven true after successful isolation of the spirochete [34]. At the time, *I. dammini* was the sole known vector. Soon, this finding was contested based on growing evidence of Lb beyond the boundaries of New England. Subsequently, it was demonstrated that *I. dammini* was not a separate species from *I. scapularis* and that the use of *I. dammini* as junior subjective of *I. scapularis* should be avoided to minimize confusion in the research and medical communities [39,40]. Later studies on other *Ixodes* species then highlighted *I. pacificus* as a western vector [41,42,43].

Once *I. scapularis* was recognized as a vector of interest, it was critical to garner more information about its underlying biology and how it impacted pathogen transmission. From these studies, it was determined that *I. scapularis* required the successful completion of a blood meal to progress through each of its life stages (larva, nymph, adult) [44,45]. During feeding, the ticks obtain the necessary nutrients for growth and survival and acquire microbes that assist in these functions [2,46,47]. This need for feeding for growth and survival is most evident during the initial stages of life. At the nymphal stage, *I. scapularis* ticks will primarily feed on small, woodland creatures like *Peromyscus leucopus*, the white-footed mouse, during which it may acquire *B. burgdorferi* (Figure 2) [16,48,49,50,51,52]. Additionally, other tick-borne pathogens may be acquired from blood meals such as *Anaplasma*, *Bartonella*, *Ehrlichia*, and *Babesia* spp. [48,53,54,55]. As *I. scapularis* nymphs molt into adults they may become more species restrictive and feed on larger prey like *Odocoileus virginianus*, the white-tailed deer, or *Ursus americanus*, the American black bear [56,57,58]. Additionally, they may also prey upon non-competent hosts such as reptiles, or on incidental hosts such as domestic animals, wildlife, and humans [59,60,61]. Historically, *I. scapularis* was most prevalent along the eastern seaboard of North America (Table 1), but with climate change its range has expanded both northward and westward.

Another hard tick species, *D. variabilis*, thrives on many hosts during all of its life stages. Like *Ixodes* ticks, this hard tick preys on small mammals early in its life cycle before parasitizing larger prey as it matures. While this vector was known as a North American anthropophilic ectoparasite, scientific interest rose with the sudden emergence of Rocky Mountain Spotted Fever outbreaks earlier in the 20th century [33,51,62]. Compounded with the precipitous upsurge of Lb, researchers initially considered this arachnid, and its close cousin *D. andersonii*, as potential vectors of the disease. In later transmission studies, however, both ectoparasites were disqualified as being capable of transmitting the etiological agents of Lb, and to date, no *Borrelia* spirochetes have been detected within *Dermacentor* spp. ticks. Due to the nature of this tick and other TBD pathogens it has been known to carry, it remains clinically relevant and is found ubiquitously throughout North America (Table 2).

Lastly, the most significant tick on the western North American seaboard is *I. pacificus* [79,80]. Similar to its east coast cousin, *I. scapularis*, *I. pacificus* is a generalist feeder that thrives on a variety of hosts (Table 1). In doing so, it can acquire and transmit a broad range of microbes, including the *Borrelial* spirochetes (Table 2) [81,82,83]. *I. pacificus* has been found to quest greater distances than other tick species during all life stages [84]. This facilitates nymph and adult *I. pacificus* to better track prey for blood meals.

As mentioned, each of the discussed tick species are vectors for multiple TBD pathogens, and their acquisition/transmission of each microbe is heavily influenced by their food supply, life stage, and environment. The community of these microbes within a metazoan is called the microbiome. The term refers to commensal, symbiotic, and pathogenic microbes that coexist in a defined space [85]. The tick microbiome includes viruses, eubacteria, archaea, and eukaryotes, and together with metazoan hosts they form a holobiome [86,87,88,89]. Until very recently, microbiome studies focused solely on the bacterial components of the microbiome due to the ease of targeting the bacterial 16S rRNA gene, which is conserved throughout the entire domain. This high degree of sequence conservation permits the identification of all species within the domain through a single PCR amplification followed by DNA sequencing. However, earlier approaches lacked specificity due to technical limitations.

Previously, many bacteria within the tick microbiome have been found critical for development, survival, the balancing of complex metabolic pathways, and the execution of numerous functions of the host they preside in [51,90]. Thus, many of the most significant microbiome exist in mutualistic relationships with their hosts. In ticks, these bacteria include members of the *Rickettsia*, *Rickettsia*-like, and *Wolbachia* genera [51,91]. Their presence/absence impacts their overall morphology, feeding habits, and retention of other microbes [51,91,92]. This is most clearly observed among tick-borne organisms like Coxiella-like genera that are essential for the survival and reproduction of *A. americanum* [93,94,95]. Similarly, *Rickettsia*-like spp. are necessary for folic acid biosynthesis within *I. scapularis* and *I. pacificus*, and the *Wolbachia* spp. for reproduction [93,94,95].

In early studies, researchers found that the tick microbiome was subject to change based on its feeding status and environmental stressors. With the advent of next-generation sequencing (NGS), the small nuances behind such changes and the impact of a given host on the microbial consortium became clearer. In these studies, *I. scapularis* was found to host a wide range of bacterial species within their microbiomes [51,90]. This complex bacterial microbiome was noted to be quite variable in terms of both breadth and diversity based on the life stage [91,96,97]. Additionally, it was noted to change based upon feeding patterns and sex [98,99]. In previous studies of *I. scapularis* adults, both male and female ticks were noted to have the *Rickettsia* and *Rickettssia*-like spp. as predominant members of their microbiome [91,96,99]. Other tick species, including *D. variabilis* and *A. americanum*, were not found to be able to sustain such a wide net of bacterial organisms [100,101,102]. Interestingly, while the breadth of the microbiome was different, a similar decrease in diversity occurred among all tick species as they progressed throughout their lifecycles [102,103,104,105].

Recently, the *I. pacificus* microbiome has also been a subject of research. Like, *I. scapularis*, it has a diverse microbiome [83,102]. However, it has yet to be determined what role its microbiome members play in the etiology of human infections. Previous efforts in characterizing the *I. pacificus* microbiome have identified common commensal bacteria including *Spiroplasma ixodetis* as well as various *Rickettsia* spp., *Rickettsia*-like spp., *Ehrlichia*-like, and *Anaplasma*-like spp. [106]. Additionally, the major human pathogen as *Borreliella burgdorferi* and *Borrelia miyamotoi* have also been found at reduced levels within this tick population [106].

While research on tick microbiomes has improved our understanding of the various TBD pathogens, there are several limitations in most studies. The first is that standard short-read sequencing in NGS systems lack the resolution necessary to identify the bacteria at the species-level. Next is the dearth of applying these technologies to elucidate the full nature of the microbial consortia within ticks. Finally, these platforms do not target microorganisms such as fungi, nematodes, and apicomplexan parasites. Thus, much remains to be elucidated regarding to the various tick microbiomes.

## 3. Borreliosis, the Most Common Type of Tick-Borne Disease

The emergence of chronic cutaneous, neurologic, arthritic, and cardiac maladies have been documented worldwide for hundreds of years [31,107,108,109,110,111,112]. In the US, these outbreaks have occurred in waves for centuries. Some of the earliest mentions were documented in Long Island during the early 1600s [113]. These cases were called ‘Montauk’s knees’, ‘Southhampton knee’, or water on the knee due to the arthritic-like symptoms [114]. In the latter half of the twentieth century, another wave occurred at Lyme, Connecticut in the beginning of the 1970s. In this small community, several children presented with numerous non-specific and arthritic-like symptoms. These ailments were later formally recognized as Lyme borreliosis and is commonly referred to as Lyme disease [107,108,109,110,111,112]. Over the years, Lb has continually increased in prevalence within the US and Canada and is now the most common tick-borne disease, accounting for ~500,000 new cases each year in the US alone—due in large part to climate change, deforestation, habitat loss, and loss of predators of the primary mammalian species upon which the ticks feed [115].

In the years that followed the discovery of Lb, its etiological agent, *Borreliella burgdorferi*, and other *Borreliella* spirochetes were found to be pathogenic to humans [38,116,117,118,119,120]. Some of these spirochetes do not, however, have the same vectors or pathogenesis. In this instance, other *Borreliaceae* members can cause Relapsing fever (Rf) and may be transmitted through a tick or louse vector. While these distinctions were made primarily on pathology and geo-locale of origin, later comparative genomic research suggested splitting the genus into two distinct disease-causing genera [121]. The Lb causing spirochetes were then given the new designation of *Borreliella*, while all other *Borreliaceae* spirochetes which cause Rf retained its original name of *Borrelia*.

Lb is a multi-systemic infectious disease with a wide and seemingly unconnected variety of conditions (e.g., polyarthralgias; neurological diseases, including polydysthesias/parathesias, cardiomyopathy, multiple sclerosis, other demylenating diseases, and ataxia; and psychiatric conditions, such as pediatric bipolar disorder and PANS and PANDAS) [122,123,124,125,126,127,128,129,130]. The sole pathognomonic presentation of Lb is erythema migrans, commonly known as the bulls-eye rash. Unfortunately, this presentation may not occur or is not visible to all individuals, occurring in approximately 50% of Lb cases [131,132,133,134]. Lb is divided into distinct stages: localized and disseminated. The disease presentations vary wildly among individuals, as well as by the species of *Borreliella*. This is illustrated most clearly with common *Borreliella* spirochetes, *B. burgdorferi*, *B.azfelii*, and *B. garinii*, each of which is endemic to the US or Europe. *B. burgdorferi*, the most common cause of Lb in the US, is primarily associated with arthritis, while *B. afzelii* is associated with cutaneous infections and *B. garinii* with neurological disease in Europe [133].

Due to the spectrum of non-specific symptoms for Lb, diagnoses are often difficult. Currently, clinicians rely on imprecise serological diagnostics and proof of tick-bite before accepting a Lb differential. While the above approaches may be useful in some instances, these current diagnostics have severe limitations, including a highly unreliable negative predictive value. To understand why these diagnostics may fail, it is critical to understand the basic biology of these spirochetes.

Previously, researchers have noted that *Borreliellal* spirochetes share many features ranging from their obligate parasitic nature within a large network of reservoir hosts, rather organisms that sustain spirochetes and facilitate their reproduction, to dynamic morphology that facilitates their near-constant host invasion [38,135]. Their unique morphology is thought to be created by 11 anti-parallel inter-membrane flagella and a chitobiose peptidoglycan [136,137]. Interestingly, this morphology has been documented to change in response to varying external stimuli [138,139,140,141,142]. It is, however, unclear what the mechanisms underlying the *Borreliaceae* morphological shifts are.

In addition to altering their morphology as a stress response, *Borreliaceace spirochetes* can manipulate their host’s immune and inflammatory response to their advantage. This is most clearly seen within *I. scapularis* ticks where *Borreliella* spp. reside within the tick midgut. These spirochetes are bound to the tick receptor for OspA (TROSPA) until the initiation of a blood meal [143]. Through feeding, the *Borreliella* dissociate from TROSPA, then switch their outer membrane surface protein (Osp) composition. The act of feeding induces tick salivary proteins to cover presenting Borreliella OspA and translocate to the tick salivary gland before peritrophic membrane formation [144,145]. Once in the salivary gland, the spirochete can then be transmitted into the new host dermis. During transmission, the *Borreliella* OspA in the outer membrane decreases and OspC rises [144]. For humans, *Borreliella/Borrelia* can be transmitted at varying rates depending on the tick species, tick feeding status, microbial strain, and microbial load, e.g., an *I. scapularis* tick can transmit *B. burgdorferi* within 24–48 h of initiating blood feeding [146,147,148].

Once *Borrelieceae* spirochetes have successfully entered the human body, they can persist within the dermis before disseminating. There are two proposed dissemination methods for *Borreliella* spirochetes: the hematogenous and non-hematogenous routes including the lymphatics or tissue [149]. In both dissemination routes, the spirochetes mitigate the host immune response to prevent recognition by the innate immune system and ultimately delay and distort the development of a T-cell-dependent B-cell response [49,150,151,152,153]. In addition, *Borreliaceae* spirochetes, can also evade the host immune system through various other means [154]. Both *Borrelieceae* can utilize the antigenic variation system, *vls*, present within the genome to evade the complement cascade. Additionally, *Borreliella* can achieve complement evasion by binding to Factor H, a negative regulator of host complement, to outer membrane proteins CspA, CspZ, and OspE [155,156,157]. *Borreliella* can also inhibit the classical complement pathway by binding C1r to outer membrane protein BBK32 [158,159]. Through antigenic switching of *Borreliella* outer membrane proteins, including hypervariable OspC and BBA70, the overall outer membrane composition and pathogenesis of the spirochete can be altered in situ [160,161]. Through these virulence mechanisms, it is believed that if *Borrelieceae* spirochetes are not successfully cleared by the immune system, they may colonize host tissues to form a persistent infection.

While Lyme borreliosis has become highly prevalent, the impact of elapsing fever (Rf) still remains a significant health concern. The RF-causing Borrelia genus can be transmitted either by ticks or lice around the world [154,162,163,164]. Within the United States, this infectious disease has remained endemic solely within western mountainous regions [164]. Regardless of geo-locale of origin, all variants of Rf have the same symptomology. The illness does have nonspecific symptoms like fatigue, headache, nausea, and muscle/joint aches [165,166]. Important diagnostically, however, is that uniquely induces periodic fever spikes associated with Borrelial septicemias. The fever dissipates during periods of time when there are decreased levels of Borrelia present within the blood but return on a cyclical basis over the course of weeks. Due to the elevated numbers of Borrelia within the blood, Rf is commonly diagnosed through microscopic examination of blood smears (Figure 3).

## 4. *Borreliaceae* Diagnostics

In the US, the cost of preventing and treating Lb has been estimated to range from $712 million to $1.3 billion per year, but this is likely a gross underestimate as many Lb patients go undiagnosed for years while seeking care for their ‘nonspecific’ symptoms [167]. Often, patients may pay out of pocket for additional diagnostic tests and treatments. In contrast, acute diagnostics for other common bacteria such as *Streptococcus*, *Treponema pallidum*, and *Staphylococcus* infections are accurate and lead to effective treatment before the bacteria can progress to later stages of infection [168,169,170,171,172]. Meanwhile, the economic impact of Lb infections continues to rise in large part due to an inadequate diagnosis. Thus, it is critical to develop and implement better diagnostics, prognostics, and therapeutics for borrelioses [7,173].

Many factors contribute to acute Lb diagnoses being missed. For example, persons with darker skin pigmentation will often not display a visible EM rash, others lack access to medical care or only have non-specific symptoms of acute Lb, while still others have a non-traditional EM rash that is not recognized during the acute phase of Lb [132,174,175]. If a patient suspected of acute Lb presents to healthcare providers, clinicians will assess risk factors for contracting Lyme borreliosis, including symptom presentation timing, geographic location, recent travel history, pet ownership, and history of other TBDs or rashes [176]. Laboratory-based tests are then utilized by clinicians to confirm a suspected acute case of Lyme borreliosis. Currently, there are several indirect and direct approaches to assist in diagnosing an individual with Lb. The Center for Disease Control and Prevention (CDC) recommends a two-tiered serological (ELISA and Western blot) system to confirm a suspected Lyme borreliosis case [177]. The two-tiered approach relies on a patient’s adaptive immune response towards transiently expressed surface proteins of *Borreliella*. Producing an IgG antibody response with strong avidity towards specific antigen targets takes between 2–3 weeks following infection [177]. Furthermore, most Lb western blots utilize *B. burgdorferi sensu stricto* strain B31 (Bb B31) as the source of the proteins utilized in their assay [178]. The B31 subtype was isolated over 30 years ago and the Bb B31 antigens do not represent other *Borreliella* antigens produced from other closely related Lb causing spirochetes [7,179]. Thus, patients who seroconvert during acute Lb infection could produce antibodies targeting antigens that are not included on the standard western blot. In addition, these serology-based diagnostic approaches cannot serve as prognostics to track treatment outcomes. This forces physicians to primarily rely on a patient’s symptoms to guide clinical outcomes or antibiotic treatment efficacy studies [180].

Serologic Lb diagnostics are further complicated by variation in the human adaptive immune response. If patients are diagnosed with Lb based on an EM rash and antibiotic treatment is promptly initiated, they might not seroconvert [181,182]. This fact further complicates the surveillance and confirmation of Lb. The lack of seroconversion could be due to Bb’s profound immunomodulatory and immunosuppressive effects which depend on the combination of host and pathogen genetics [152,183]. Accurate diagnosis of Lb is further complicated when a patient is co-infected with other *Borreliellal* spp. or additional tick-borne disease pathogens that are also commonly transferred from the tick’s mid-gut [7,184,185,186,187,188]. Taken together, the average sensitivity of the Lb two-tiered test for the acute Lb is less than 50% [178]. This poor sensitivity produces high rates of false-negatives and delays treatment which contributes to the development of chronic/late-stage Lb. During late-stage Lb, such as Lyme carditis or Lyme arthritis, a two-tiered test can confirm the diagnosis of the patient with high sensitivity. Unfortunately, patients in the later stages of Lb face permanent tissue damage and require longer antibiotic treatments [189,190].

Newer serologically-based diagnostics present recombinantly expressed surface proteins from multiple pathogens and strains of *Borreliella* [185,191]. These approaches increase the chance of detecting antibodies produced towards *B. burgdorferi* strains other than B31 or identifying co-infections. These methods, however, still have their limitations, as Bb is immunosuppressive and, depending on the infecting strain and host genetics, a significant percentage of infected persons will fail to appropriately produce antibodies.

Clinicians seeking diagnosis for suspected Lb patients may venture beyond CDC guidelines. Traditional pathologic assays such as dark-field microscopy and primary culture from blood or skin biopsies have poor sensitivity and are not employed as a reliable diagnostic for Lb, however they have a very high positive predictive value [192]. Attempting to culture or detect *Borreliella* spirochetes from human tissue biopsies using PCR methods (standard or quantitative) is invasive and insensitive [7,193,194,195,196,197]. These direct methods are limited by the low spirochete load in tissues and the bloodstream, unlike many other human bacterial pathologies [7,185,186,198,199,200]. Thus, patients and clinicians require alternative methods of acute Lb diagnosis.

In contrast to the limitations of above-mentioned PCR-based methods for detecting *Borreliella* DNA within humans, Next-Generation Sequencing (NGS) approaches can be highly specific. As *Borreliella* spirochetes are rarely present in the blood after initial disease onset, the challenge for NGS is to obtain enough of a sample to confidently detect genes associated with acute Lb bacteria [8,201]. Previously, many *Borreliella* NGS approaches targeted highly conserved genes throughout the genera such as the ribosomal 16S in ticks or human samples [202]. However, this approach was limited to identifying Lb within ticks rather than humans due to low titers of *Borreliella* in the bloodstream.

In a new NGS approach, the limitations of the sample sources may be circumvented by using patient urine [203]. While most NGS-based assays are limited by the low counts of *Borreliella* genomic material present in human samples such as blood, this approach aims to ensure a higher *Borreliella* DNA yield with claims that Lb bacteria infect the kidneys. While accuracy was stated to be ‘superior’ to the standard two-tiered testing approach, the sensitivity of the test has yet to be reported in the literature. This is slightly different than previous diagnostic iterations which used the same biosample but targeted solely OspA, a protein which would not be expressed in high quantities on the outersurface of *Borreliaceae* within a mammalian host [204].

Other efforts have been made to increase the sensitivity of Lb bloodborne detection. Traditional PCR-based diagnostics for Lyme borreliosis have been improved by isothermal amplification of DNA, followed by PCR amplification of *Borreliellal* DNA, which is then detected by electrospray ionization mass spectrometry (PCR/ESI-MSI). In PCR/ESI-MSI, it was possible to detect the presence of *B. burgdorferi* in 13 of 21 blood samples from patients with an acute Lb cases confirmed with positive serology and a history of at least one EM rash [8,200]. The assay required 1.25 mL of EDTA-treated whole blood and could detect 0.6 or greater copies of *Borreliella* genomes in whole blood.

In a follow-up study, the PCR/ESI-MSI method attempted to survey the presence of *B. burgdorferi sensu stricto* within four patients during their antibiotic treatment for Lyme borreliosis [205]. In this study, the investigators increased the blood volume from 1.25 mL to 20 mL, with the aim of increasing the diagnostic sensitivity. *B. burgdorferi* genes were detected in 2 of the 4 patients acutely infected with the aforementioned spirochete. In addition, they did not determine if the increased blood volume increased sensitivity. The genomic amplification approach relied on detecting and targeting conserved genes present within the *Borreliella* genome such as rpoC, FlaB, and OspC [206]. While such targets can indicate the presence of this spirochetal genus and may provide species-level resolution, there are some complications [205]. One such complication is that use of consistently expressed proteins like OspC may be insufficient due to the protein’s high diversity.

Another NGS diagnostic approach utilizes unbiased metagenomic cell-free DNA sequencing of human plasma. This cell-free DNA (cfDNA) approach was used to detect *B. burgdorferi* DNA from 64% (18 of 28) human plasma samples during *acute* Lb [207]. The cfDNA sequencing method’s sensitivity was further improved by combining the results of the modified two-tiered serology testing to identify 86% of acute Lyme borreliosis cases. Additionally, a recent NGS detection study identified core genes within the *Borreliella* pangenome to increase the sensitivity of DNA-based *Borreliaceae* diagnostics [208]. Taken together, genomic *Borreliella* detection methods have significantly improved over the last decade. However, more work is required to deliver a robust and sensitive diagnosis for patients and clinicians. One possibility is to use innate immune proteins that recognize specific PAMPs, such as Apolipoprotein H linked to paramagnetic beads, to ‘sweep’ a much larger volume of blood [209,210,211,212,213].

Next-generation sequencing has been used to detect host responses to acute Lb rather than attempting to directly detect the *Borreliellal* genome. Sequencing human T-cell receptors (TCRs) is a novel approach to Lyme disease diagnostics and began clinical trials in 2021 [214]. T-cells respond to Lb infection earlier than B-cells can produce antibodies, and thus the expansion of *Borreliellal*-specific T-cell receptor sequences in a patient’s circulating lymphocytes has the potential to confirm acute cases of Lyme disease earlier than traditional serologic methods [151]. This TCR immuno-sequencing assay differentiated acute Lb patients from healthy controls with a sensitivity of 54%, while the standard two-tiered serological testing approach had a sensitivity of only 30%. Clearly, human T-cell responses significantly vary between patients and this approach will not detect all acute Lyme disease cases. However, TCR immuno-sequencing assay’s increased sensitivity is a move in the right direction and has the potential to be combined with other diagnostic approaches to further increase sensitivity.

In addition to genomic detection, researchers have explored xenodiagnosis, metabolomics, and biomarker profiling [215,216,217]. Xenodiagnostics use an uninfected, natural vector for the isolation of the targeted pathogen from the infected host [218]. In the case of *Borreliellal* spirochetes, ticks facilitate the reacquisition of *Borreliella* from a variety of hosts during a 24-h feeding cycle [219,220]. This feature was noted in the past with Lyme-infected monkeys and mice but, was not substantiated in humans until 2014 [221,222,223]. Over the last 8 years, a clinical trial of tick-based recapture of *Borreliellal* pathogens from infected human hosts has been underway [218]. No results from this study have been released at this time. This approach could be further bolstered by applying NGS to characterize the pathogens recaptured after the tick feeds on the patient suspected of contracting Lyme borreliosis to increase diagnostic sensitivity. It is important to note, that while it could prove useful, much more work would be necessary to make it feasible as a diagnostic.

Metabolomic analyses of Lyme borreliosis patients has also made great strides in recent years. These studies have identified altered abundances of circulating metabolites produced by host tissues during Lyme borreliosis. A recent assay was able to discriminate between acute Lyme borreliosis and uninfected controls using their metabolic profiles [224,225]. Diagnostics relying on specific metabolic profiles are limited by the time and cost associated to prepare samples for analysis but offer yet another promising avenue for future Lyme borreliosis diagnostics.

Proteomic studies of serum collected from humans afflicted with Lb by Zhou et al. identified host acute phase protein abundance alterations during acute Lyme borreliosis [217]. The abundance of proteins—APOA4, C9, CRP, CST6, PGLYRP2, and S100A9—were validated using a second sample set of acute Lyme borreliosis samples and discriminated between healthy controls and acute Lb patients with a 78% sensitivity. Developing a multiplexed ELISA to identify acute-phase proteins associated with Lyme borreliosis could yield a high-throughput diagnostic, yet the issue of cross-reactivity with other infection markers must first be addressed. “Mimic diseases” such as rheumatoid arthritis or fibromyalgia often have similar acute phase protein alterations. Thus, the Lb proteomic study should be validated for Lb-specificity by testing against a panel of sera from patients with other mimic diseases [224].

Lastly, a glycoproteomic approach using MALDI-FT-ICR mass spectroscopy has been demonstrated to detect changes in the IgG N-glycan profile during acute Lyme disease with a sensitivity of 75% and specificity of 100%. Moreover, this assay can differentiate between acute Lb cases and patients who have received successful doxycycline treatment [226].

In summary, *Borreliella* diagnostics are improving, but have a long way to go as each has strengths, as well as limitations [178]. In this era of increasing TBDs, the best path forward may be to combine multiple diagnostics to complement the strengths of each method to construct a testing protocol that is highly sensitive and specific. In doing so, better measures can be taken to initiate early treatments and prevent chronic disease progression.

## 5. The *Borreliella* and *Borrelia* Genomes

This near-constant cycle of host transmission, acquisition, and host immune evasion suggests that all *Borreliaceae* spirochetes would contain a large, complex pan-genome. This hypothesis was challenged when the first *B. burgdorferi sensu stricto* genome was sequenced in the late 1990s [227]. It showed that *B. burgdorferi* held a small singular linear chromosome and several linear and circular episomes and prophages. These findings were echoed in later sequencing projects in other *Borreliella* spp. Additionally, early *B. burgdorferi* genomic studies, revealed many housekeeping genes were contained on the chromosome, progressing from the center [227,228]. Within them were genes for ribosomal machinery, 32 transfer (t)RNAs, and tandem repeats. Surprisingly, genes necessary for de novo biosynthesis of carbohydrates, amino acids, nucleosides, and lipids were not among them. *Borreliella* also lacked genes for iron-requiring proteins. In their place were genes for several kinds of scavenger proteins or proteins that relied on manganese, magnesium, or zinc for cofactors [229,230,231,232,233,234,235]. This absence of genes for key enzymes was not unique among obligate pathogens. In later years, these stark absences were mirrored within the chromosomes of many Lb spirochetes, to the point where each species varied by less than 8% in terms of gene content [66,117,119,236,237,238,239,240].

While similar studies for Rf spirochetes have been conducted, less is known about their genome and gene content. The first *Borrelia* genome was sequenced in 2008 by Lescot et al. [241]. Since then, researchers have found that *Borrelia* spirochetes, like *Borreliella* spp., have a singular linear chromosome and many plasmids [209,242,243,244]. The *Borrelia* chromosome is also similar in its GC skew and contains many housekeeping genes [241,245]. It does vary in that it has fewer copies for the 5S and 23S rRNA genes in comparison to *Borreliella* chromosomes [241]. Additionally, the *Borrelia* chromosome still retains many of the genes critical for purine salvaging and glycerol biosynthesis which is suspected of facilitating survival and rapid growth [244]. Others have found a high degree of synteny within the chromosome amongst *Borrelia* species with less than 2% difference, or the absence of one gene [244]. Currently, a quarter of the chromosomal genes remain uncharacterized.

As mentioned, both the *Borreliella* and *Borrelia* spirochetes have a large plasmid repertoire. *Borreliella* spp. more so than *Borrelia* spp. The gene content within the *Borreliellal* plasmids is sparse. The few identifiable genes contained within are unique to an individual species that could assist in host adaptation [246]. The distribution of these virulence genes can be on either the small linear or circular replicons. To date, the *Borrelial* linear plasmids (lp) have been suggested to contain fewer functional genes and are laced with pseudogenes [247,248,249]. In contrast, the circular plasmids hold more coding genes [250]. It is possible that the lp-associated pseudogene set serves as part of an evolutionary mechanism to provide for the rapid emergence of new functional genes through recombinational processes, or that they play roles in gene regulation [250,251]. This can be seen in different studies such as those conducted by Dulebohn et al. [252]. In one instance the *Borreliella* lp28-3 investigated, and it was found that several genes, though considered as non-essential to *Borreliella* survival or as pseudogenes, facilitated spirochete survival throughout the infectious cycle [252].

Previous *Borreliella* genomic studies have found plasmids to contain a significant fraction of the genes necessary for spirochetal adaptation, survival, or pathogenesis. Most of these plasmids are under the stringent control and are present as single copies within the genome—making the *Borreliellal* genome, in this sense, more similar to multi-replicon genomes of eukaryotes. The classification of these episomal elements has been fraught with multiple difficulties as the genes are largely unannotated and can only be identified via paralogous family (Pfam) groups, leading to classifications based on size in kilobases and topology. Current plasmid naming conventions do not have any relation to actual biological functions. Within the *Borreliella* genome, three plasmids are thought to be the most relevant physiologically: linear plasmid 54, linear plasmid 17, and circular plasmid (cp) 26.

Lp54 is uniform in gene density and without repeat elements. The genes present on this plasmid create surface localized lipoproteins that interact directly with host immunomodulators necessary for tick transmission and mammalian infection. The smaller episome, lp17, differs from lp54 by containing genes for homologous proteins with unknown functions and an end with high homology to other linear *Borrleiella* replicons. Lp17 also varies greatly among different *Borreliaceae* species, amounting to 16 variants identified to date [247]. Lastly, cp26 is highly syntenic and contains genes that encode key metabolic and nutrient import proteins. Interestingly, cp26 also contains the *OspC* genes critical for mammalian infection and the resolvase T enzyme which has been documented as necessary for genome replication [253,254]. As stated, most of the Lb plasmids are present as single copies except for cp32 [249]. This replicon is highly syntenic among various species and strains and carries numerous *osp* genes necessary for survival in murine, and tick hosts. Genes needed for successful host adaptation and immune evasion are also found in the lp28 replicon including the has the VLS antigenic variation system [247,255].

*Borrelia* spirochetes contain fewer, but slightly larger, plasmids. Within these replicons are genes for metabolism, replicon maintenance, and pathogenesis [244]. *Borrelia*, similar to their Lyme borreliosis cousins, have both linear and circular episomes. The naming of Rf episomes also follows similar conventions. RF linear plasmids are divided into two categories based on size. The first group of linear plasmids is composed of replicons that are approximately 200 Kb in size. The second group is much smaller, averaging at 30 Kb, much like those seen in Lb spirochetes. These smaller plasmids encode genes utilized for antigenic variation which provide RF spirochetes the capacity to switch between antigenic variants, quickly lending to immune evasion, and explaining the ‘relapsing fever’ phenotype [244]. It is important to note that there is one known RF cp that is syntenic with Lb cp32 [256]. The contents of the replicons are also primarily for the generation of surface proteins.

## 6. Pangenomic Applications

Considering the genomic differences among the *Borrelieace* spirochetes and their pathogenic heterogeneity, there are understandably profound therapeutic differences. As the incidence and prevalence of borrelioses rise, it is imperative to develop new approaches to diagnostics. Recently, a new field called pangenomics has provided researchers with a large toolset of comparative genomic technologies to characterize compositional, functional, and structural differences among large numbers of genomes at any taxonomic level. This has been exemplified by recent pangenomic analyses of the pathogenic bacterial species Clostridioides *difficile*, *Escherichia coli*, *Haemophilus influenzae*, and *Moraxella catarrhalis* [257,258,259,260,261,262,263,264,265,266,267]. These studies have helped identify new targets for the development of vaccines, diagnostics, and therapeutics [257,258,259,260,261]. Genus-level pangenomes have also been described [257,261,268,269,270,271,272].

In the case of *Borreliella/Borrelia*, discerning key features of the genome has been difficult due to biological features and technical limitations. Borrelia’s complex genome includes repetitive sequences, covalently-bound hairpin ends (akin to telomeres), and a multitude of linear and circular plasmids. These challenges were difficult to overcome until the advent of next-generation sequencing. Recently, various Illumina platforms have been used due to their technological improvements, allowing for the sequencing of Borrelia’s many plasmids. Unfortunately, while this is a step forward in *Borreliellal/Borrelial* genomics, such platforms are still ill-suited for properly capturing the large stretches of repetitive sequences in some of these episomes [273]. Thus, establishing near-complete genomes was exceedingly challenging without the use of additional platforms.

This drove Borrelial geneticists to search for ways to capture the entire genome in all of its complexity. The arrival of the latest models of Pacific Bioscience’s long-read sequencing platforms that utilize the high-fidelity circular consensus sequencing protocols has allowed researchers to attain much longer and more accurate sequence reads. This method has been applied to both Lb and Rf spirochetes [66,242]. There were concerns, however, about capturing smaller episomes within the spirochete’s genome using this technology.

Concurrently with the growth of *Borreliellal/Borrelial* genomics, a tentative preliminary pangenome, rather defined as the complete collection of all genes available within a species (or higher taxonomic unit), from existing sequenced and annotated *Borrelia* genomes was created. This research focused on *B. burgforferi sensu stricto* and the *B. burgdorferi sensu lato* complexes [208,274]. In this first pangenome study, several Lb spirochetes were chosen based on their host of origin, geo-locale origin, vector, and if they had been isolated from symptomatic individuals. These 22 *Borreliella* genomes were then used to construct genome-wide, single-nucleotide-polymorphism phylogenies [274]. The researchers checked the pangenome size for *B. burgdorferi* alone and at the genus-species level for the entire *Borreliella burgdorferi sensu lato* complex. It was suggested that *B. burgdorferi* by itself had a ‘closed’ pangenome. This finding suggests that there were a limited number of novel additions of new genes added to the genome for each *B. burgdorferi* genome used in the analysis. Further expansion of the pangenome project to encompass other *Borreliella* members revealed a different pangenome composition. It was shown that as a genus, *Borreliella* had an open pangenome that would continue to expand with each new genome addition. While this study advanced the field of *Borreliellal* comparative genomics and pangenomics, there were limitations brought primarily from the technical capabilities of sequencing platforms, which were prevalent at the time of research.

Subsequent studies on the *B. burgdorferi* pangenome were smaller in scale. One study centered on the design and application of a potential xenodiagnostic through the use of the pangenome by targeting a fragment of the *B. burgdorferi* 16S rRNA gene [208]. This was then applied with limited success on *I. ricinius* ticks [208]. This could stem from the limited sequences of the *Borreliella* 16S gene. Additionally, this gene has been documented to exist as a single copy within the *Borreliellal* genome, which is commensurate with its slow growth and long doubling times.

In the age of long read, next-generation sequencing, it is possible to overcome many of the afore-mentioned limitations. With the recent expansion of PacBio sequencing platforms, a *B. burgdorferi*, *Borreliella burgdorferi sensu lato* complex being constructed as well as establishing the first pangenome for the *Borreliaceae* family. Thus, novel biomarkers can be identified as diagnostic targets at the species and genus levels. It is anticipated that this new and expanded pangenome for the *Borreliaceae* family will yield more accurate diagnostics, new antibiotic targets specific to *Borreliella*, and improve outcomes for patients diagnosed with *Borreliella*-based infections.

## Figures and Tables

**Figure 1 genes-13-01604-f001:**
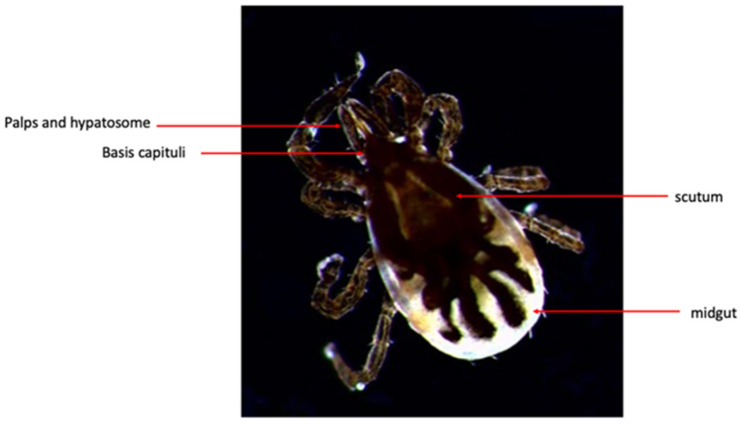
The anatomy of the adult *I. scapularis* hard tick underneath a dissection microscope. The red arrows indicate the different anatomical parts of the *I. scapularis*.

**Figure 2 genes-13-01604-f002:**
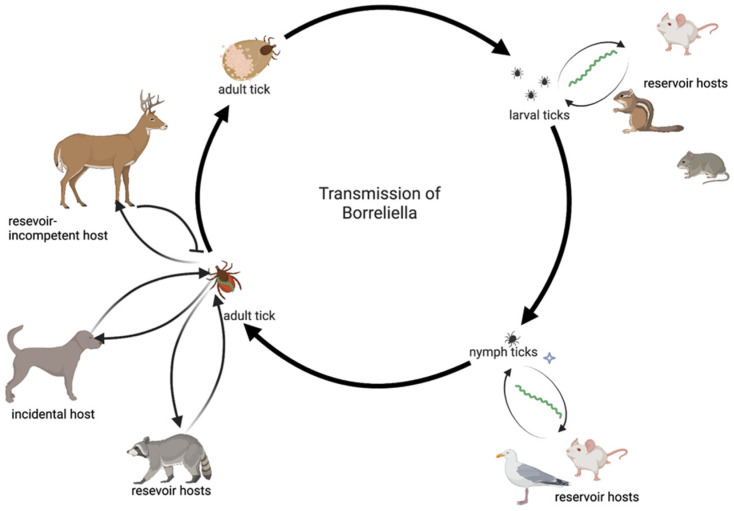
Transmission and acquisition of *Borreliellal* spirochetes. The image was created in BioRender by KM Socarras 2022.

**Figure 3 genes-13-01604-f003:**
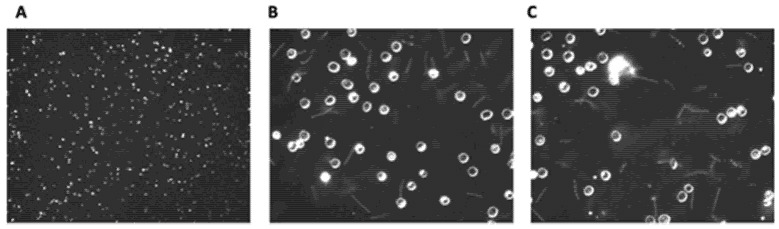
Dark field microscopic images of wet mount Borrelial spirochaetes (*Borrelia hermsii* strain DAH). Blood from C3H-Hej mice infected with *B. hermsii* was diluted 4-fold with phosphate buffered saline. (**A**): 100×, (**B**,**C**): 400×.

**Table 1 genes-13-01604-t001:** Leading anthropophilic ticks within the United States.

	*Amblyomma americanum*	*Amblyomma maculatum*	*Dermacentor andersonii*	*Dermacentor variabilis*	*Ixodes cookei*	*Ixodes pacificus*	*Ixodes scapularis*	*Rhicephalus sanguineus*	*Ornithodoros hermsii*
**Common name** ^1^	Lone Star Tick	Gulf Coast Tick	Rocky Mountain Wood Tick	American Dog Tick	Groundhog Tick	Western Blacklegged tick	Blacklegged Tick/Deer Tick	Brown Dog Tick	
**Type** ^3^	Hard	Hard	Hard	Hard	Hard	Hard	Hard	Hard	Soft
**Prey** ^1,3^	rodents, domestic animals	rodents, domestic animals	rodents, domestic animals	domestic animals	groundhogs, skunks, rodents, racoons, foxes, weasels, domestic animals	mice, voles, weasels, deer, birds, lizards	mice, deer, birds	domestic animals	rodents
**Feeding style** ^3^	3-host tick	3-host tick	3-host tick	3-host tick	3-host tick	3-host	3-host	3-host tick	
**Dimorphic** ^3^	Yes	Yes	Yes	Yes	Yes	Yes	Yes	Yes	Yes
**Zone** ^1,2^	East and Southern US	South-mid US, Southern Arizona		East of Rocky Mountains, US, pacific coast	Eastern U.S	Western US	Eastern, Upper Midwest, and Southern US	World-wide, * Southern border of U.S	Northwest coast, Texas, Florida
**Peak activity** ^1,2^	Early Spring- Late fall	Early Spring- Late fall		Spring and Summer	Early spring-Mid Fall	Early spring-Mid Fall	Early spring-Mid Fall		Year-round
**States** ^1,2^	AL, AR, CT, DE, FL, GA, IL, IN, IA, KS, KY, LA, ME, MD, MA, MS, MO, NE, NH, NJ, NY, NC, OH, OK, PA, RI, SC, TN, TX, VT, VA, WV, DC	AL, AR, FL, GA, KS, LA, MS, MO, NC, OK, SC, TN, TX, VA	AZ, CA, CO, ID, KS, MT NE, NV, NM, ND, OK, OR, SD, UT, WA, WY	AL, AR, CA, CT, DE, FL, GA, IL, IN, IA, KS, KY, LA, ME, MD, MA, MI, MN, MS, MO, MT, NE NH, NJ, NY, NC, ND, OH, OK, PA, RI, SC, SD, TN, TX, VT, VA, WV, WI, WY, DC	ME	AZ, CA, NV, OR, UT, WA	AL, AR, CT, DE, FL, GA, IL, IN, IA, KS, KY, LA, ME, MD, MA, MI, MN, MS, MO, NH, NJ, NY, NC, ND, OH, OK, PA, RI, SC, SD, TN, TX, VT, VA, WV, WI, DC	AL, AK, AZ, AR, CA, CO, CT, DE, FL, GA, HI, ID, IL, IN, IA, KS, KY, LA, ME, MD MA, MI, MN, MS, MO, MT, NE, NV, NH, NJ, NM, NY, NC, ND, OH, OK, OR, PA. RI, SC, SD, TN, TX, UT, VT, VA, WA, WV, WI, WY, DC	
**Habitat** ^1,2^	Wooded areas	Coastal areas	scrublands, lightly wooded areas, and open grasslands	Human settlements				Human settlements	

^1^ Center for Disease Control and Prevention 2022. ^2^ National Environmental Health Association 2022. ^3^ Soneshine, Daniel E. (1992). Biology of Ticks Volume I. Oxford University Press. * This tick species is found within the United States and several countries within Europe.

**Table 2 genes-13-01604-t002:** Leading tick-borne pathogens within the United States and their vectors.

	*Babesia microti*	*Anaplasma phagocytophilum*	*Borrelia burgdorferi*	*Borrelia miyamotoi*	*Borrelia mayonii*	*Ehrlichia chafeensis*	*Ehrlichia muris*	*Ehrlichia ewingii*	*Francisella tularensis*	*Rickettsia parkeri*	*Rickettsia rickettsii*	*Bourbon virus*	*Coltivirus*	*Heartland virus*	*Powassan virus*
**Type**	Parasite	Bacteria	Bacteria	Bacteria	Bacteria	Bacteria	Bacteria	Bacteria	Bacteria	Bacteria	Bacteria	Virus	Virus	Virus	Virus
**Disease Name** ^1^	Babesiosis	Anaplasmosis	Lyme Borreliosis	Relapsing Fever	Lyme borreliosis	Ehrlichiosis	Ehrlichiosis	Ehrlichiosis	Tularemia	*R. parkeri* spotted fever	Rocky Mountain Spotted Fever	Bourbon virus disease	Colorado Tick Fever	Heartland virus disease	Powassan virus disease
**Discovery** ^1^	1990 [63]	1932 [64]	1982	1995 [65]	2016 [66]	1986 [67]	2009 [68]	1996	1912 [69]	1937 [70]	1906	2014 [71]	1950 [72]	2012 [73]	1990 [74]
**Reservoir** ^1^	Small mammals: [55] *Peromyscus leucopus*, *Procyon lotor*, *Blarina brevicauda*, *Tamias striatus*	*Peromyscus leucopus*, *Odocoileus virginianus*	*Peromyscus leucopus*, *Odocoileus virginianus*, *Tamias striatus*, *Blarina brevicauda*, *Sorex cinereus*, *Sciurus carolinensis*, etc. [53]	*Peromyscus leucopus*, *Apodemus* spp., *Microtus* spp., *Tamias* spp., *Sciuridae* spp., etc. [75]	*Peromyscus leucopus*, *Tamiasciurus hudsonicus* [76]	*Odocoileus virginianus*	*Peromyscus leucopus*	*Odocoileus virginianus*	Rodents	Unknown					Deer
**Tick****Vector** ^1^	*I. scapularis*	*I. scapularis*	*I. scapularis*	*I. scapularis*	*I. pacificus*	*A. americanum*	*I. scapularis*	*A. americanum*	*A.Americanum*, *D. variabilis*, *D. andersonii*	*D. andersonii*	*D. variabilis*, *R. sanguineus*, *D. andersonii*	*A. americanum*	*D. andersonii*	*A. americanum*	*I. scapularis*, *I. cookei*

^1^ Center for Disease Control and Prevention 2022. In contrast, the lone star tick, *A. americanum*, is found only in select regions in North America. This arachnid has a notably aggressive feeding behavior towards prey during all life stages. It targets primarily large prey such as *O. virginianus*, but also domestic animals and humans. While doing so, it can transmit several tick-borne pathogens such as *Ehrlichia ewingii* and *E. chaffeensis* (Table 2) [67,77]. To date, this tick has not been documented as capable of transmitting *Borrelial* spirochetes and as such is not considered a Lb vector [78].

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
