# Peer review of "Large-Scale Sequencing of Borreliaceae for the Construction of Pan-Genomic-Based Diagnostics"

_genes, 2022, doi:10.3390/genes13091604_

Round 1

Reviewer 1 Report

In this review paper the authors provide a historc overview of the process to developing the pan-genomic-based diagnostics for borreliaceae.

I do not have any major objections related to this manuscript and I have listed some minor suggestions in the PDF of the manuscript (please see the attached file), e.g. to use the term "Lyme borreliosis" when applying to disease and to to use borreliae or any other or more specific name when speaking about the causative infectious agents.

I am not a native English speaker, but it seems that the small/big capital letters are not used uniformly as well as the abbreviations, e.g. Lb.

The authors mention that erythema migrans is the pathognomonic sign of early Lyme borreliosis. I also propose to mention that in the majority of clinical cases of erythema migrans (which is the most prevalent clinical manifestation of Lyme borreliosis), no additional diganostics, beyond clinical exam and epidemiologic data is needed. 

Reviewer 2 Report

There are no line numbers in this manuscript which has made things a bit difficult while referring to particular statement for comments. I expect this review to be a bit more focused to subject, thoroughness and attention to detail.   In Section 1: Vectors and their microbiome, heading does not justify the content fully. More details need to be provided to establish a correlation between tick vectors, endosymbionts and tick-borne pathogens. It could not establish a link between Borreliaceae and pan-genomic-based diagnostics. Please improve it so that it can justify the title of the paper.    More details should be provided for Figure 3. There is no mention about Borrelia hermsii  in the text. Please discuss its relevance if you mention it in the figure.

Please correct Rhicephalus annaltus to Rhipicephalus annulatus.   Please provide a little more details about tick Ixodes uriae (which spirochete it vectors)
  In Page 4 - "Once I. scapularis was recognized as a vector of interest, it was critical to garner more information about its underlying biology and how it impacted pathogen transmission. From these studies, it was determined that I. scapularis required the successful completion of a blood meal to progress through each of each of its life stages (larva, nymph, adult)" 
Comment - On page 4, please remove one of 'each of'. 
  On page 6, "The tick microbiome includes viruses, eubacteria, archaea, and eukaryotes and together with metazoan hosts they from a holobiome". Please correct it as 'form'.
